# Recent Advances in Functionalized Electrospun Membranes for Periodontal Regeneration

**DOI:** 10.3390/pharmaceutics15122725

**Published:** 2023-12-04

**Authors:** Luana Epicoco, Rebecca Pellegrino, Marta Madaghiele, Marco Friuli, Laura Giannotti, Benedetta Di Chiara Stanca, Andrea Palermo, Luisa Siculella, Vuk Savkovic, Christian Demitri, Paola Nitti

**Affiliations:** 1Department of Engineering for Innovation, University of Salento, 73100 Lecce, Italy; rebecca.pellegrino@unisalento.it (R.P.); marta.madaghiele@unisalento.it (M.M.); marco.friuli@unisalento.it (M.F.); christian.demitri@unisalento.it (C.D.); 2Institute of Medical Physics and Biophysics, University of Leipzig, 04103 Leipzig, Germany; 3Department of Biological and Environmental Sciences and Technologies, University of Salento, 73100 Lecce, Italy; laura.giannotti@unisalento.it (L.G.); benedetta.dichiara@unisalento.it (B.D.C.S.); luisa.siculella@unisalento.it (L.S.); 4Implant Dentistry College of Medicine and Dentistry, Birmingham B4 6BN, UK; andrea.palermo2004@libero.it; 5Clinic and Polyclinic for Oral and Maxillofacial Plastic Surgery, University Hospital Leipzig, 04103 Leipzig, Germany; vuk.savkovic@medizin.uni-leipzig.de

**Keywords:** periodontal disease, guided tissue regeneration (GTR), electrospun membranes

## Abstract

Periodontitis is a global, multifaceted, chronic inflammatory disease caused by bacterial microorganisms and an exaggerated host immune response that not only leads to the destruction of the periodontal apparatus but may also aggravate or promote the development of other systemic diseases. The periodontium is composed of four different tissues (alveolar bone, cementum, gingiva, and periodontal ligament) and various non-surgical and surgical therapies have been used to restore its normal function. However, due to the etiology of the disease and the heterogeneous nature of the periodontium components, complete regeneration is still a challenge. In this context, guided tissue/bone regeneration strategies in the field of tissue engineering and regenerative medicine have gained more and more interest, having as a goal the complete restoration of the periodontium and its functions. In particular, the use of electrospun nanofibrous scaffolds has emerged as an effective strategy to achieve this goal due to their ability to mimic the extracellular matrix and simultaneously exert antimicrobial, anti-inflammatory and regenerative activities. This review provides an overview of periodontal regeneration using electrospun membranes, highlighting the use of these nanofibrous scaffolds as delivery systems for bioactive molecules and drugs and their functionalization to promote periodontal regeneration.

## 1. Introduction

The relationship between microbial communities and the host is essential for maintaining the body’s homeostasis, therefore changes in the composition of the human microbiota can lead to severe dysregulation of body functions [1].

The human microbiota is a collection of trillions of tiny organisms, represented by bacteria, archaea, viruses, phages and fungi [2]. They inhabit various anatomical sites within our bodies, and their composition is generally distinct among individuals. The microbiome is recognized as a second genome that has a symbiotic relationship with the host and is actively involved in regulating its metabolism [3]. Microbiota ecosystems evolve with the host’s chronological development, modulating early physiological development, nutrition, immunity and pathogen resistance at all ages [4].

Considering the oral cavity, the oral microbiome is often thought of as a single entity; but in reality, the oral cavity comprises several niche-specific microbiomes, including those of the tooth surfaces, gums, saliva, tongue, buccal mucosa, palate, subgingival and supragingival plaque, pharynx and tonsils [5].

In this context, a shift in the structure and composition of the bacterial community, known as microbial dysbiosis causes disruption of the structure responsible for ensuring tooth attachment and stability, leading to a disease known as periodontitis [6]. Like other infectious diseases, it is characterized by persistent inflammation, with immune cell activation leading to the release of pro-inflammatory cytokines and the recruitment of phagocytes and lymphocytes [7]. This carries a risk for the development of life-threatening co-morbidities.

Periodontal therapy is selected according to the severity of the disease, ranging from treatment with antibiotics and probiotics in the early stages of the disease to surgical treatment in the advanced stages [8]. In the latter case, the goal is the complete restoration of both the architecture and function of the periodontium, resulting in alveolar bone and cementum regeneration, new connective periodontal ligaments (PDL) and resolution of the high state of inflammation [9]. In dentistry, different strategies are needed to achieve this goal, as the aim is to restore lost or damaged periodontal tissues, that differ in cellular and biochemical composition [10].

Among these, the techniques of tissue/bone grafting and guided tissue/bone regeneration (GTR/GBR) in the field of tissue engineering (TE) are of great importance and in constant development.

In recent years, nanotechnology has made a significant breakthrough in the field of periodontal regeneration. Nanofibers, in particular, have become popular due to their outstanding properties, which make them unique compared to other nanomaterials (e.g., nanotubes and nanoparticles), such as high surface-to-volume ratio, flexibility and high porosity [11].

Several technological methods have been used to produce functional nanofibers, such as self-assembly polymerization, melt blowing, template blowing and solution blowing [12]. Among these, electrospinning (ES), an effective and versatile technology in which polymer nanofibers are extruded using an electrohydrodynamic process, is one of the leading methods, as it is characterized by its simplicity of use, while at the same time guaranteeing the production of continuous nanofibers with different desirable structures on the nanometric or microscopic scale, with excellent properties. In addition, compared to their bulk material, electrospun nanofibers generally appear to have superior mechanical properties [13]. 

This technique has attracted considerable attention in a wide range of applications, including the fabrication of bioactive GTR/GBR scaffolds [14]. Indeed, due to their specific micropatterning, characterized by a high surface area to volume ratio and controlled porosity with interconnected pores, electrospun nanofibers have an excellent ability to mimic the arrangement of a native extracellular matrix (ECM). This allows for enhanced protein uptake, activation of specific gene expression and intracellular signaling [15]. By varying the ES parameters, extensive mesh-like networks or aligned mats can be obtained. The topography of electrospun mats plays an important role in regulating cell activities, from spreading and orientation to proliferation, migration and differentiation [16]. Several studies have reported an improvement in the mechanical response of aligned fibers compared to random fibers. In particular, aligned scaffolds are useful for regenerating tissues with preferential orientation such as PDL [16]. 

Various natural polymers such as collagen, hyaluronic acid (HA) and synthetic polymers such as poly(vinyl alcohol) (PVA) and poly(glycolic acid) (PGA) have been used. In order to combine their advantages and attempt to reduce their disadvantages, such combinations of polymers (blend polymer solutions) have been widely produced. The following sections provide a description of some of the most commonly used polymers. 

In addition, specific manufacturing conditions and post-processing arrangements could be tailored to obtain membranes with specific functions and active materials to accelerate and increase the efficacy of tissue regeneration [17]. The functionalization with pharmacological compounds [18], metallic, metallic oxide and carbon-based nanoparticles have been adopted to increase the outcome of these scaffolds [19].

In this work, a concise description of the periodontium and periodontal disease will be presented, followed by a summary of some of the current strategies used for the production and functionalization of electrospun nanofibers for the achievement of periodontal regeneration.

## 2. Complex Architecture of the Periodontium

The periodontium is a sophisticated and functional assembly of multiple tissues that encompass and provide essential support to the tooth. This complex unit comprises four distinct tissue types, each differing in terms of location, biochemical characteristics, cellular composition, protein types and quantities, mineralization, metabolic activity levels, and susceptibility to diseases. It comprises two hard connective tissues, represented by cementum and alveolar bone, and two soft tissues, namely the gingiva and the PDL [20]. Due to its heterogeneous composition, the periodontium serves specific functions such as upholding the tooth’s stability, shielding it from oral microflora, and enabling the tooth to attach securely to the bone [21]. 

### 2.1. Gingiva: Composition of Periodontal Tissues

The gingiva, commonly known as gums, constitutes a component of the masticatory lining within the oral mucosa. It encases the external surface of the periodontal unit, which includes the alveolar processes of the jaws, and enfolds the neck of the teeth. This configuration forms an effective protective barrier for the underlying periodontal tissues [22]. The gingiva consists of a stratified squamous epithelium, which is predominantly cellular in nature, and a central core of connective tissue in which collagen fibers are prevalent [23]. Anatomically it is made of three delimited regions named marginal, attached, and interdental gingiva. 

Marginal (or free or unattached gingiva) is the free end of the gingiva, characterized by its smooth surface. It surrounds the neck of the tooth and defines the gingival sulcus, and in about half of the cases, the gingival sulcus, which is a shallow linear depression, distinguishes this area from the adjacent attached gingiva [24]. 

The interdental zone of gingiva is non-keratinized and located in the area between the two adjacent teeth beneath the contact point [25]. 

The attached gingiva is stippled, firm and strongly attached to the periosteum of underlying alveolar bone and to the cervical area of the tooth by means of junctional epithelium located in the floor of gingival sulcus [26].

### 2.2. Periodontal Ligament (PDL): The Tooth’s Anchor and Guardian

The PDL is an aligned fibrous connective tissue that connects the tooth root to the alveolar bone. It is firmly anchored to the root cementum of the teeth at one end and to the alveolar bone of the jaw at the other, thus supporting the teeth in situ within the jawbone. In addition, it performs many essential functions: (1) contributes to nutrition and tissue homeostasis and repair of damaged tissues; (2) provides mechanical stability; (3) through the neural network and the presence of mechanosensors, regulates the sensory input of the masticatory system, protecting the tooth and alveolar bone from possible high forces; (4) forms a protective barrier, together with the gingiva, against pathogens from the oral cavity [27].

The PDL is composed of diverse cell populations, with fibroblasts being the predominant type, responsible for the formation and maintenance of the ECM, which is rich in proteins and glycosaminoglycans. Other cell components include endothelial cells, epithelial cells of Malassez, cells associated with the sensory system, and cells with the potential to differentiate into either cementoblasts or osteoblasts [28,29]. This diverse composition, combined with its high vascularity and the presence of bundles of collagen fibers, confers unique properties on the PDL. These properties include the ability to rapidly adapt to multidirectional masticatory forces and a significant capacity for self-renewal and repair [28]. The blood vessels of the PDL provide the nutrients, immune cells, growth factors and hormones that are essential for remodeling the surrounding periodontal tissues [30]. It is involved in proprioception by relaying information to the somatosensory cortex, which is crucial for avoiding occlusal forces during activities such as mastication and speaking [31]. 

### 2.3. Alveolar Bone: Tooth Socket and Protector

Alveolar bone (or alveolar process) is a mineralized connective tissue that forms the tooth socket in which the root part of the tooth is anchored, protecting it and providing masticatory functions [32]. Histologically, the alveolar bone consists of a thin layer of compact bone that is adjacent to the PDL, the lamina dura, which lines a central core of trabecular bone. Specifically, the PDL pierces through the lamina dura and anchors at one side to the alveolar bone embedded within this bone are the extrinsic collagen fiber bundles of PDL (Sharpey’s fibers), with the other end connected to the cementum [33]. Alveolar bone remodels like non-oral skeletal sites but at a rate three to six times higher [34]. Under homeostatic conditions, osteoclasts, which resorb old bone matrix, and osteoblasts, which subsequently produce new bone matrix, operate at the same rate. When the osteoclast and osteoblast functions are imbalanced, with osteoclast activity overriding, this turnover does not take place properly, resulting in bone loss [35]. 

### 2.4. Cementum: Tooth Support and Connection

Cementum is a mineralized, avascular connective tissue that covers the entire dentin surface and is considered part of both the tooth structure and the periodontium because of its supporting role. In collaboration with the PDL and the alveolar bone, cementum contributes to the establishment of a support network for the tooth. This network establishes a connection between the tooth and the alveolar bone through the calcified collagen fibers known as Sharpey’s fibrils. These fibers are crucial for detecting and responding to occlusal mechanical pressure during biting and chewing [36]. Cementum can be broadly classified into two primary parts: acellular and cellular cementum. The acellular cementum, situated near the tooth’s crown, is characterized by densely packed mineralized tissue and contains relatively few embedded cementocytes. In contrast, cellular cementum, located toward the tooth’s apex, harbors a greater population of cementocytes and displays a lesser degree of mineralization [37].

Our bodies harbor huge numbers of microbial cells, organized into complex communities that specifically inhabit different niches of the human body, such as the skin and the respiratory, gastrointestinal, oral and urogenital tracts. Such ecosystems perform a wide range of functions that are essential for the host’s well-being [38]. The human oral cavity hosts an abundant and diverse microbial community, called oral microbiome (OM), playing a key role in periodontal health [5]. Periodontitis is caused by the overgrowth of commensal microorganisms that grow in biofilms, unlike many infectious diseases that are caused by infection with exogenous pathogens. 

## 3. Periodontitis Onset and Progression

### 3.1. Stages of Periodontitis

The normal periodontium provides the support needed to maintain the teeth in a healthy state; consequently, its impairment gives rise to various issues known as periodontitis. The term periodontitis refers to a group of chronic multifactorial inflammatory diseases mediated by an interaction between dysbiotic microbial communities and aberrant immune responses, affecting the periodontal tissues [39], (Figure 1). The disease can be broadly divided into four stages based on severity: gingivitis (early periodontitis), mild, moderate and severe periodontitis. This classification is not a permanent body, but rather evolves with the development of new knowledge. The possibility to categorize the different stages of periodontitis enables the design of effective treatment for the patient. Indeed, if detected in time, it may be treated successfully, while if left untreated causes the destruction of the periodontal structure and it is considered to be the main cause of tooth loss, which affects about 20–50% of global population with prevalence increasing with factors such as age, gender (males > females) and lower socio-economic status.

Recently, the World Health Organization estimated that 5–20% of middle-aged (35–44 years) adults in Europe, and up to 40% of older people (65–74 years) have periodontitis [40]. The mildest form of periodontal disease, gingivitis, is an inflammation of the gingiva induced by plaque accumulation and accompanied by redness, bleeding and edema [41]. Bacteria penetrate deep into the surrounding periodontal tissues: cells on the tooth root surface and cementum are destroyed, and the oral mucosal epithelium grows downward, producing a gingival crevice. Bacteria take up residence in this cleft, and the ensuing inflammatory process can cause the breakdown of periodontal tissues, including cementum, periodontal ligament and alveolar bone.

### 3.2. Microbial Factors in Periodontitis

Gram-negative anaerobic bacteria spirochetes and viruses are the main factors in the disease [42]. Over the years it has been established that the healthy state is dominated by the bacterial genera *Streptococcus*, *Fusobacterium, Actinomyces* and *Corynebacterium,* whereas the diseased state is mainly dominated by pathogenic genera such as *Prevotella*, *Treponema* and *Fusobacterium*. Although there are some gaps in the knowledge of the principal pathogenic bacteria, the decreased abundance of *Proteobacteria* and *Actinobacteria* and the increased abundance of *Bacteroidetes* and *Firmicutes* in periodontitis has been confirmed by several studies [43]. Alveolar bone loss can in turn lead to the formation of a periodontal pocket around the tooth, which serves as a reservoir for the growth of anaerobic bacteria (e.g., *Porphyromonas gingivalis* and *Prevotella intermedia*) [44]. 

The local inflammatory response, involving both innate and acquired immunity, can evolve into systemic inflammation and metabolic changes associated with an alteration of the gut microbiota and beyond [45]. The taxonomic composition of the OM associated with severe periodontitis revealed the presence of the so-called “red complex” bacteria, including *Porphyromonas gingivalis, Tannerella forsythia*, and *Treponema denticola*. High densities of bacteria belonging to the phylum *Synergistetes, Firmicutes,* and *Bacteroidetes* were also found. While taxa such as *Fusobacterium nucleatum*, *Veillonella parvula* and some types *of Streptococcus* sp., often showed the same relative abundance in both healthy and diseased states [46].

### 3.3. Systemic Implications of Periodontitis

Nowadays, there is an increasing body of scientific literature reporting an epidemiological association with other chronic inflammation-driven disorders, with multiple mutual connections between them. These systemic inflammatory diseases include diabetes, cardiovascular diseases, respiratory infection, gastrointestinal and colorectal cancer, neurological diseases (Alzheimer’s disease) and adverse pregnancy outcomes, although all the mechanisms behind this association have not been fully elucidated. In contrast, the role of systemic host pathologies, such as autoimmune disorders and diabetes, in periodontal disease pathogenesis has been well noted [47].

The hypothesis of the status of systemic low-grade inflammation (LGI) as a common background of several diseases including periodontitis, is under increasing discussion, indeed LGI could be a silent risk factor for many of them (Figure 2). Following this concept, periodontitis might contribute, at least in part, to the development and progression of chronic systemic diseases by persistently inducing a condition of LGI [48]. However, it is not only the taxonomic composition of the OM, but the overall metabolic activity in the oral habitat, including the host response to microbial products, which are the critical factors distinguishing between oral health and dysbiosis. Bacteria also directly contribute to systemic inflammation through the release of toxins or other microbial products into the bloodstream. For this reason, the evaluation of proteome and/or metabolome could provide a more accurate discrimination between healthy and unhealthy state [49].

On the other hand, the role of oral virome (bacteriophages, viruses and retroviruses) in the disease development has not been yet largely investigated. In this regard, periodontal disease has emerged as a risk factor for COVID-19 disease [50]. Few studies have identified the ability of the virus to replicate directly in oral tissues; in particular, Huang et al. recently identified 34 unique cellular subpopulations within the gums and salivary glands that harbor SARS-CoV-2 viral entry factors. Indeed, SARS-CoV-2 uses the host receptors ACE2 and TMPRSS to enter cells, and techniques such as single-cell RNA sequencing and fluorescence in situ hybridization have demonstrated the expression of these two types of receptors in salivary glands [51].

Considering the potential repercussions of this disease, the adoption of proper oral hygiene practices, professional care and remotion of risk factors, such as smoking, and unequilibrated diet, when coupled with proficient diagnostic and therapeutic approaches, may have the capacity not only to prevent the onset of periodontitis but also to exert a beneficial influence on the occurrence and severity of severe systemic diseases beyond oral health [52]. 

## 4. The Innovation of Tissue Engineering and Regenerative Medicine in Periodontal Therapy

Bearing in mind the nature of the disease, the selection of appropriate therapy poses a formidable challenge. Depending on its severity, various curative approaches may be adopted, ranging from non-surgical therapy to surgical therapy in more advanced stages. Non-surgical treatments were shown to decrease pocket depth and enable the formation of new dental attachments in the early stages of periodontitis, with scaling and root planning recognized as the preferred approach. Several studies have demonstrated that to achieve greater success, it is preferable to combine additional therapies such as antimicrobial and antibiotic drugs or host immunomodulatory agents [18]. 

Advanced stages of periodontal disease necessitate the resolution of a deeper periodontal pocket. In these cases, traditional treatments involve the removal of damaged and/or inflamed tissues without the repairing and regeneration of the lost one [53]. In this field, TE and Regenerative Medicine (RM) have provided new approaches that can aid in the restoration of structure and function of the damaged tissue. 

### 4.1. Tissue Engineering (TE) and Regenerative Medicine (RM)

TE and RM have gained more and more interest in recent decades and continue to expand fast in clinical fields [54]. In particular, TE and RM have the same origin and goals: TE is described as an interdisciplinary field that applies engineering and life science principles and methods to understand the structure and function of normal and pathological mammalian tissues and try to restore, maintain, or improve tissue function [55]; and also RM applies engineering and life sciences principles to heal or replace damaged tissues, promote organ renewal, and try to restore lost functions [56]. Although they are both based on the idea that new tissues can be generated from living cells seeded on a scaffold [57], TE is an ex vivo procedure in which cells are seeded into a scaffold, giving them physical, chemical and mechanical stimuli in order to generate a fully tissue that can be later implanted in the affected tissue, while RM is an in vivo procedure, in which a biomaterial scaffold with or without seeded cells is directly implanted into the body to facilitate regeneration of the defected area [58]. Nowadays, the two disciplines are known together as TERM [58] and create a multidisciplinary field of research [59], from life sciences to biology and engineering [60]. They are based on three key factors: scaffolds, cells and regulatory signals [61].

Scaffolds are 3D frameworks that are needed to support cell growth and proliferation during tissue regeneration [62]. They are made of several kinds of biomaterials, a class of materials “designed to take a form that can direct, through interactions with living systems, the course of any therapeutic or diagnostic procedure” [63]. In this sense, biomaterials can regulate molecular signals and cellular behaviors to achieve tissue regeneration even if they are not loaded with cells or drugs [64]. They are characterized by unique chemical, mechanical and biological properties; for example, in this case, osteoinductivity and osteoconductivity, which make them suitable and safe to interact with living tissues [65]. They can have a natural or synthetic origin and have the role of inducing partial or full tissue replacement [66]. In this regard, biomaterials and, consequently, scaffolds have to be biocompatible; thus, they should not be toxic to cells. On the contrary, they preferably should interact with them [67]; they should be porous in order to allow cells to enter and colonize, with a proper pore size that allows nutrients and waste exchange [68]. Scaffolds should also be mechanically similar to the tissue that has to regenerate to prevent the overgrowth of surrounding tissues and maintain its structure [69]. Finally, they should be biodegradable, with a degradation rate that matches the regeneration rate of the target tissue [70].

Cellular sources represent the most important challenge in this field. The best method is to harvest autologous cells directly from the patient to avoid immune response following transplantation, but this method presents major limitations, such as donor-site morbidity, limited expansion in vitro and difficulty in obtaining suitable cells in case of old patients, that are almost prohibitive for this approach [71]. The alternative is the use of cells from human or non-human donors. In this case, in addition to donor morbidity, the risk of transmitting infectious agents to the patients and ethical problems, discourage this method [72]. For all these reasons, stem cells seem to hold great potential, as they are commonly found in many tissues, they can renew themselves and differentiate into many phenotypes [73]. They can also be used when they are not fully differentiated and multipotent, and they can be altered to avoid immune response in patients [74]. 

Regulatory signals are of fundamental importance because cells always require signals from the environment to proliferate and differentiate in a specific way [75]. These external signals can be generated through mechanical–chemical and physical stimuli as well as growth factors and cell–cell and cell–ECM interactions [76]. 

### 4.2. TE and RM in Periodontal Therapy

In the case of periodontal disease, for a long time, bone grafts have been largely employed in this field, filling the empty space originating from damaged tissue and allowing the regeneration of new ones acting as a biocompatible and biodegradable scaffold, releasing growth factors and supporting cell growth with good osteoconductive and osteoinductive properties [77].

At present, a wide variety of bone grafts, such as autograft, demineralized freeze-dried allograft and bovine xenograft, are used in the clinical setting for the repair of alveolar bone defects [78]. Unfortunately, most of them lack some basic characteristics; for example, ceramic ones are usually fragile and cannot be re-formed during clinical application to correctly fill the gap of the lost tissues, while injectable hydrogels can overcome these limits but with poor mechanical properties and fast degradation that lead to implantation failure. On the other hand, dissolving ceramic compounds can provide ions (Ca, P, Si and Zn) to stimulate osteogenic cell activity [79].

In recent years, guided regeneration therapy has gained more interest because it is based on the idea that cells with regenerative capabilities can aid in the regeneration of lost tissue [80]. For periodontal disease, two surgical approaches are used: GTR and GBR. They consist of surgical techniques used in dentistry and periodontics to promote the regeneration of periodontal tissues, including bone and soft tissues, that have been lost due to periodontal disease or other factors [81]. These techniques are employed to restore and maintain the health and functionality of the supporting structures of teeth. In particular, GBR is specifically aimed at promoting the regeneration of lost alveolar bone around teeth or for implant site preparation [82]. It is commonly used in cases where there is insufficient bone volume to support dental implants or when bone loss has occurred due to trauma, infection or tooth extraction. GBR employs barrier membranes, bone graft materials (e.g., autografts, allografts, xenografts or synthetic materials), and sometimes various growth factors (Figure 3). The barrier membrane prevents soft tissue infiltration and allows space for bone cells to regenerate, while the bone graft material provides a scaffold for new bone formation [83].

GTR is focused on the regeneration of periodontal ligaments, cementum and gingival tissue. The procedure involves the use of a barrier membrane, usually made of biocompatible materials, which is placed between the soft tissue and the defect site [84]. The membrane acts as a physical barrier to prevent the migration of unwanted cells from the surrounding soft tissue into the defect, allowing periodontal cells to repopulate the area and regenerate lost tissue. 

The GTR membranes are designed to resemble the scale and morphology of the ECM, creating the space necessary for the injured tissue to regenerate and, thanks to the small pore size of the membranes, preventing fibroblast invasion that would otherwise inhibit regeneration [85,86]. 

Given the unique characteristics of the oral environment, a periodontal barrier membrane designed for tissue regeneration needs to satisfy particular standards: (1) biocompatibility to ensure cell attachment and proliferation; (2) absence of cytotoxicity and immunogenicity to prevent any adverse reactions or rejection by the body; (3) a high degree of porosity to stimulate tissue growth and vascularization, thereby promoting protein absorption and preventing the overgrowth of epithelial and connective tissue [87]; (4) specific mechanical properties to withstand the mechanical stresses of surgery and the surrounding tissues in the surgical field [88] and so on.

Various methods, including the use of cast membranes, ES and dynamic filtration, have been used by researchers to produce different types of periodontal barrier membranes [89].

Both GTR and GBR techniques may be used separately or in combination, depending on the specific clinical situation. The success of these procedures depends on various factors, including the patient’s overall health, the surgeon’s skill, the quality of the barrier membrane and graft material, and proper post-operative care. These techniques have significantly improved the possibilities for saving teeth and achieving successful dental implant placement in patients with compromised periodontal and bone support. However, their success also relies on patient compliance with post-operative instructions, regular follow-up appointments, and good oral hygiene practices.

A further important distinction in maxillofacial surgery is between the use of resorbable and non-resorbable membranes [90]. Resorbable membranes gradually dissolve in the body over a period of time, typically a few weeks to a few months. This natural degradation eliminates the need for follow-up surgery to remove the membrane, reducing patient discomfort and cost. Whereas, non-resorbable membranes require additional surgery for their removal [91]. The former are made of both naturals and synthetics polymers such as collagen [92], chitosan [93,94,95], poly(lacticco-glycolic acid) [96]. They must present a slow degradation in order to persist in the site for at least 4 to 6 weeks, the time needed to have an effective tissue regeneration [97]. In particular, bioresorbable membranes made of collagen, are employed, due to their significant functional, and morphological potential which provide the opportunity to examine cellular behavior, as confirmed by structural and ultrastructural studies [98]. The latter are made from materials like expanded polytetrafluoroethylene (ePTFE) or titanium-reinforced PTFE mesh and are often used in more intricate cases where long-term support is necessary to prevent soft tissue from collapsing into a healing site [99,100].

In the next paragraphs, an overview of electrospun membranes for GTR and their functionalization will be provided.

## 5. Electrospinning Technique

The ECM is a dynamic and complex fibrous network of proteins and polysaccharides that provides structural support to cells and mediates their functions. ES, by offering the possibility of obtaining scaffolds that mimic the natural ECM, has become a widely used technique in the field of RM and TE [101].

The basic setup for ES (Figure 4) is fairly simple and consists of four main components: a volumetric syringe pump containing a polymer solution, a metal needle (spinneret), a power supply and a static or dynamic metal manifold with variable morphology [102]. 

The syringe containing the polymer solution is connected to a spinneret through which a constant and controllable flow of solution can be supplied. A high potential difference is applied between the syringe capillary and the spinneret and an electro-hydrodynamic process forms an extruded polymer droplet at the tip of the needle [103]. When the suspended droplet of polymer solution is subjected to an electric field strong enough to overcome its surface tension, the droplet is then exposed to the repulsive force between its surface charges and the Coulomb force exerted by the external electric field, causing conical distortion of the droplet, commonly referred to as a ‘Taylor cone’, from which a charged jet emerges [104]. The jet initially travels in a straight line, then undergoes violent jerky movements due to bending instabilities, during which the solvent evaporates, leading to the continuous formation of nanofibers [105].

The selection of solvent is critical to ensure proper spinning of the polymer. Solvents such as acetone, ethanol, hexafluoro isopropanol, chloroform, dichloromethane and dimethylformamide are widely used, but some are expensive and harmful to the environment. In addition, traces of organic solvents remaining in electrospun materials are detrimental to biomedical applications such as tissue engineering. Water can be utilized as a solvent to overcome this problem when using hydrophilic polymers [13].

There are several parameters that affect morphology and fiber diameter, which can be classified into spinning parameters, solution characteristics and environmental factors [106]. By optimizing these parameters and selecting natural and/or synthetic polymers (Table 1), solvents and adding specific bioactive molecules, it is possible to tailor the electrospun mats completely to the final application.

## 6. Functionalization of Electrospun Membranes to Potentiate Periodontal Regeneration

Designing a membrane that guarantees a harmonious equilibrium between its physical and mechanical attributes while simultaneously meeting biological standards is imperative and poses a substantial challenge. To achieve this goal, various bioactive materials, including proteins, growth factors, drugs, bio-ceramic and functional polymeric substances, can be integrated into the “third generation” of nanofibrous membranes, through surface coatings or biomolecule incorporation. In this manner, these matrices not only act as a barrier, but also regulate the biological behavior of cells, the local inflammatory microenvironment, angiogenic, osteogenic properties and regenerative capabilities [115]. For the production of drug-loaded GTR membranes, the ES technique has gained increasing attention. Molecules, such as antibiotics and anti-inflammatory drugs, are often used as adjuvant therapy to improve the outcome of the treatment. Nanoscale drug delivery technologies have attracted considerable attention for the advantages they offer over traditional systemic drug delivery. These include controlled release over time, improved patient compliance and reduced side effects and toxicity. In fact, nanomaterials should be metabolized into non-toxic particles easily eliminated through the bloodstream [116]. The mechanism and kinetics of drug release from biodegradable materials occur through three basic phases, respectively, initial burst release, diffusion-controlled release and degradation-controlled release. In the first phase, a massive release of drug, which had been localized on the fiber surface during the ES process, occurs; followed by the second phase in which the release is controlled by diffusion phenomena and the third phase in which molecules are also released due to the degradation of the polymer network [117,118]. Additionally, the intrinsic parameters of the protein delivery process (solvent exposure and electrical field) may affect the 3D structure, rendering a protein inactive or eventually toxic. 

Coaxial ES has been investigated to overcome these problems. It involves a modification of the traditional (blend) ES process in which two different solutions (core and sheath materials) are pumped through nozzles, resulting in a core-sheath fiber morphology. The biomolecule solution forms the inner jet, and is co-electrospun with a solution forming the outer jet [119]. 

A similar core-sheath structure can be obtained by emulsion ES, which involves the simultaneous spinning of two immiscible solutions (Figure 5). The biodegradable fiber-forming polymer is dissolved in organic solvent to form the continuous phase, while the active ingredients are dissolved in aqueous solutions to form the aqueous phase. During ES, the continuous phase evaporates rapidly, resulting in an increase in viscosity. As a result, the droplets of the aqueous phase containing the drug migrate towards the center of the jet due to the viscosity gradient [120].

Compared to coaxial, it can still damage biomolecules due to the interface tension between the aqueous and the organic phases of the emulsion [17].

### 6.1. Anti-Infective Drugs

The addition of an antibacterial agent has become one of the most commonly used strategies, as bacterial infiltration is the main reason for the failure of GTR/GBR membranes in clinical applications. Antimicrobials and antibiotics can be administered locally, allowing a slow-release system to deliver antibacterial drugs to the site of infection over an extended period of time, eliminating the toxic side effects of long-term systemic administration. Their efficacy in enchaining the clinical outcomes can be demonstrated by evaluating parameters such as reduction in probing depth and increase in clinical attachment level (CAL) [121,122].

Metronidazole (MNZ) is currently the mainstay of treatment for anaerobic infection, which is the main form of infection in periodontitis. For example, Jiajia Xue and colleagues developed anti-infective GTR membranes by combining polylactic micro/nanofibers with gelatin. This formulation resulted in enhanced biocompatibility and biodegradability; they also loaded a wide range of MNA contents (1–40 wt%). In experiments involving subcutaneous implantation in rabbits, the MNA-loaded membranes (with the optimal drug concentration found to be 30% MNA) induced a milder inflammatory response compared to pure Polycaprolactone (PCL) nanofibers, by inhibiting the growth of anaerobic bacteria [123]. Rui Shi et al. realized a biocompatible membrane for infection-sensitive GTR based on electrospun PCL. Through a multi-step reaction, they bound MNA to the membrane via ester bonds. The ester bonds can be selectively hydrolyzed by cholesterol esterase, an enzyme secreted by macrophagocytes that accumulate at the site of infection and whose concentration is positively correlated with the severity of infection. This dynamic system seems promising in the GTR antibacterial membrane field [124].

Due to their effective bacteriostatic activity against Gram-positive and Gram-negative bacteria, tetracycline antibiotics and their derivatives (e.g., doxycycline (DOX)) are widely used in the treatment of periodontitis and peri-implantitis [125]. Studies in rats have demonstrated the anti-inflammatory properties of DOX, a promising candidate for use in bone repair in several pathologies, including periodontal disease; the researchers provided evidence that DOX inhibits the Dkk-1 pathway while activating Wnt signaling, which is involved in the differentiation of osteoprogenitors into osteoblasts [126].

In another study, the antibiotic azithromycin was loaded into a GBR membrane made from electrospun medical-grade PCL (mPCL) fibers using a solvent evaporation technique, allowing controlled release of azithromycin and inhibiting the growth of *Staphylococcus aureus*. In addition, they showed that membrane implantation into a rodent calvarial defect polarized macrophages towards the M2 phenotype and enhanced bone regeneration, promising craniofacial and orthopedic applications [127].

### 6.2. Immune Response Modulation

Besides microbial dysbiosis, the host inflammatory response is central to this multifactorial disease and several anti-inflammatory agents have been investigated for use in periodontal therapy [128]. Non-steroidal anti-inflammatory drugs (NSAIDs), such as ibuprofen (IBU) and aspirin, can inhibit cyclooxygenase activity, thereby blocking the conversion of arachidonic acid to the prostaglandin PG, which can adversely affect periodontal regeneration [129]. NSAIDs are used to treat inflammation in rheumatic, osteoarthritic and other conditions because of their analgesic, antipyretic and anti-inflammatory properties. However, because some non-specific cytotoxic effects of NSAIDs may be harmful to the human body, further research is needed [130].

As an alternative to NSAIDs, innovative compounds known as resolvins are attracting a great deal of attention for their ability to control the duration of inflammation without stopping it, which is essential for the contraction of infection. These compounds include omega-3 fatty acids and docosahexaenoic acid [131].

### 6.3. Other Bioactive Agents and Innovative Approaches

Functional growth factors, hormones and proteins can be added to regulate the inflammatory microenvironment or optimize osteogenesis activity. Since bone repair involves the orchestral coordination of different natural bioactive components, it would be desirable to be able to incorporate some of them into membranes [132], but this presents a major challenge in terms of releasing the individual biomolecules at the most appropriate time. Growth factors such as bone morphogenetic proteins (BMPs), platelet-derived growth factor (PDGF) and simvastatin have been shown to promote tissue regeneration. Molecular interactions between growth factors and other bioactive signals should be explored before the use of combinations of these biomolecules. For example, Muthukuru et al. combined BMPs, which are directly involved in bone formation, with DOX, which is known to indirectly promote hard tissue regeneration. They evaluated biomarkers such as alkaline phosphatase (ALP), a relatively early biochemical marker of osteoblast differentiation [133], and found that the synergistic effect of the two components results in lower regeneration than when used individually [134]. 

Gene delivery has traditionally been carried out using viral or non-viral vectors. The former are associated with some risks of immune response and potential genomic integration and carcinogenic outcome [135], while non-viral vectors have lower transfection potency and shorter expression duration [136]. To overcome these drawbacks, a gene-activated matrix (GAM) has been proposed, which combines gene delivery on engineered scaffolds [137]. In this regard, ES has gained greater consensus among other suitable scaffolding techniques, offering the advantage of a spatiotemporal release of genetic material in the defective area. On the other hand, the plasmid carrying the gene of interest is inevitably exposed to the organic solvents used in the standard electrospinning technique, which may affect its integrity. To overcome this problem, coaxial electrospinning was chosen instead of the conventional single ES, resulting in innovative and effective GAM electrospun scaffolds [138,139].

It should be noted that the use of bioactive proteins has been limited by the short half-life of the protein, potential denaturation and high cost, which has prompted research into the use of bioactive nanomaterials [140]. Nanomaterials, such as ceramics and metals, have attracted increasing attention due to their outstanding biomimetic and physicochemical properties [141]. For example, nano-hydroxyapatite (nHA) is used to enhance bone regeneration and has osteoconductive and osteoinductive properties [142]. Likewise, bioglasses (BG), a large group of materials consisting of silicon dioxide, sodium oxide, calcium oxide and phosphorus pentoxide, are widely used in bone regeneration studies due to their ability to form a reactive carbonate HA layer, but also to regenerate hard and soft tissues due to their ability to bind to both types of tissues [143]. The introduction of antimicrobial nanoparticles such as silver and zinc oxide has emerged as an effective solution against a wide range of multi-drug resistant pathogens, as bacteria have developed drug resistance over time, making some infectious diseases challenging to cure [144]. Silver nanoparticles can be produced in a controlled manner, with controlled size and morphology and using them in combination with antibiotic enhance their bactericidal activity. These nanoparticles have also been reported to have an anti-inflammatory effect through modulation of cytokine and growth factor levels [145]. 

Likewise, gold nanoparticles (AuNPs) can regulate inflammation and cytokine production, resulting in increased newly formed periodontal attachment, bone and cementum [146]. Zhang et al. reported for the first time the size-dependent effects of AuNPs on the osteogenic differentiation of PDLPs and found that 45 nm AuNPs could exhibit significant anti-inflammatory effects. Their results showed that AuNPs affected the osteogenic differentiation of PDLPs in a size-dependent manner with autophagy as a possible explanation, suggesting that AuNPs with defined size could be a promising material for periodontal bone regeneration [147].

Some examples of compounds that have been successfully incorporated into electrospun membranes are shown in Table 2.

### 6.4. Multilayered Electrospun Mats

The need for multi-structured membranes arises from the complex anatomy of the natural periodontium. Potentially, multiphasic GTR membranes/scaffolds with different physical structures (e.g., pore size distribution, mechanical strength, etc.) and chemical composition (e.g., mineral components, bioactive factors, etc.) can co-ordinate hard and soft tissue responses during the defect healing process to ensure multi-tissue regeneration [81], (Figure 6). Sequential spinning or multi-layering may be employed to fabricate multilayered membranes, or alternatively, ES can be combined with other technologies like tri-dimensional printing scaffolds [156], freeze-drying or phase separation processes [91].

Recently, electrospun polylactic acid (PLA)-poly-caprolactone (PCL) nanofibrous membranes with freeze-dried chitosan and microporous channel structure were prepared by combining electrospinning, directional freeze-drying and cross-linking reaction. In vitro and in vivo tests showed that these membranes were able to stimulate the repair of periodontal defects by promoting the formation of alveolar bone, periodontal ligament and cementum-like tissue [157].

## 7. Conclusions and Future Prospective

The complexity of the oral environment makes its regeneration a challenging task. Although various natural and synthetic biomaterials have been used and modified with bioactive components, the efficacy for complete periodontal regenerative endodontics is still difficult to achieve.

ES is the most popular technique and fabrication of nanofibers due to its simplicity, affordability, flexibility and ability to spin a wide range of polymers. The encapsulation of numerous drugs and other bioactive molecules in electrospun fibers has been successfully accomplished, but some limitations related to the controlled release of biological active substances remain to be overcome. Indeed, the harmonization of ratio and release times of biomolecules is of fundamental importance. Additionally, the preparation of electrospun fibers involves organic solvents that may pose a risk to humans if used in the biological field. Therefore, in the future, various post-processing treatments will need to be addressed to avoid the possible release of residual solvents that could have a strong impact on human health, while at the same time improving the use of non-toxic solvents.

The biocompatibility and efficacy of these scaffolds are tested using in vitro monolayer or two-dimensional (2D) cell cultures, which are clearly not fully representative of the complex cellular, chemical and physical interactions that occur in organs and therefore individuals [158]. An improvement in their characterization could therefore be provided by three-dimensional (3D) tissue constructs, in which the interactions between different cell types, including stem cells and immune cells, and mediators may be faithfully reproduced [159]. Not only that, but it could also allow us to reduce the number of animal experiments, which are currently mainly based on rats [160], rabbits [161] and pigs [162].

In recent years, biological barrier membranes have gained increasing importance in periodontal regeneration and have become established in clinical practice. Commercially available membranes include non-absorbable membranes such as expanded polytetrafluoroethylene ((e-PTFE, Gore-Tex^®^ (W.L. Gore & Associates, Flagstaff, AZ, USA)), titanium-reinforced high-density polytetrafluoroethylene (Ti-d-PTFE) and bioabsorbable membranes such as collagen-based ((Bio-Gide^®^ (Geistlich Biomaterials, Wolhusen, Switzerland), Biomend^®^ (Zimmer Biomet, Inc., Carlsbad, CA, USA), etc.) [163]. 

In contrast, very few clinical reports have been published for electrospun membranes. For example, in a study conducted in 2019, 15 patients were implanted with either a commercial product (Epi-Guide^®^, Kensey Nash Corp., Exton, PA, USA) or the ES membrane made of PLA and β-tricalcium phosphate (ES 95/β-TCP). Radiographs and various indices such as the clinical attachment level were measured six months before and after surgery and showed a significant regenerative outcome in both cases. This study showed that the ES PLA95/β-TCP membrane can be used as an alternative GTR membrane for clinical applications, although no statistically significant differences were found [164].

In conclusion, the application of electrospun materials in the dental field is very promising; however, clinical and in vivo studies are lacking, and the properties of the different membranes need to be more accurately defined. To achieve this, extensive interdisciplinary research is essential to design electrospun membranes with high regenerative potential and properties tailored to the specific needs of the patient. 

In this regard, it can be summarized that the progress made over time in the fight against this complex disease is impressive, but the future prospects are still vast and need to be explored with new interdisciplinary approaches that will allow the transition from basic research to clinical application.

## Figures and Tables

**Figure 1 pharmaceutics-15-02725-f001:**
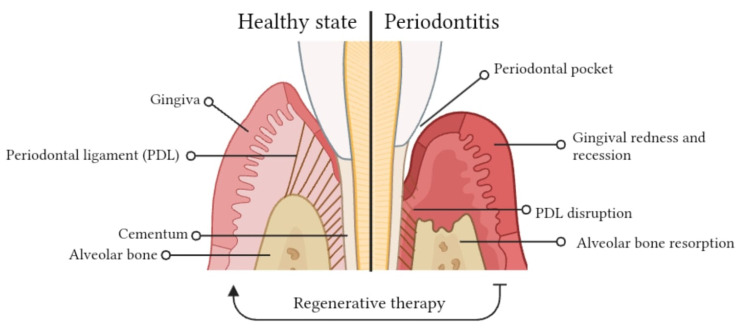
Schematic illustration of healthy and diseased periodontal structures.

**Figure 2 pharmaceutics-15-02725-f002:**
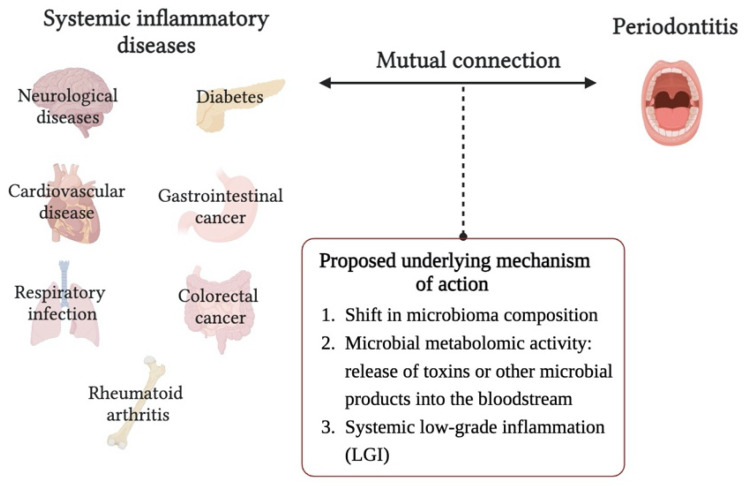
Illustration of the interrelationship between periodontitis and some systemic inflammatory diseases.

**Figure 3 pharmaceutics-15-02725-f003:**
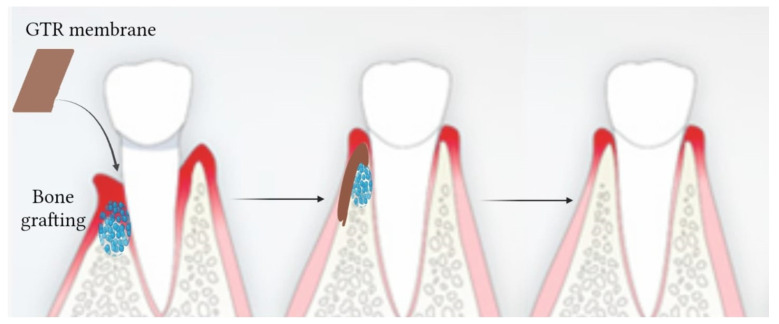
Schematic illustration of GTR membrane combined with bone grafting. The GTR membrane is placed on top of the graft material that fills the deficient bone site for the time necessary to rebuild the damaged tissue.

**Figure 4 pharmaceutics-15-02725-f004:**
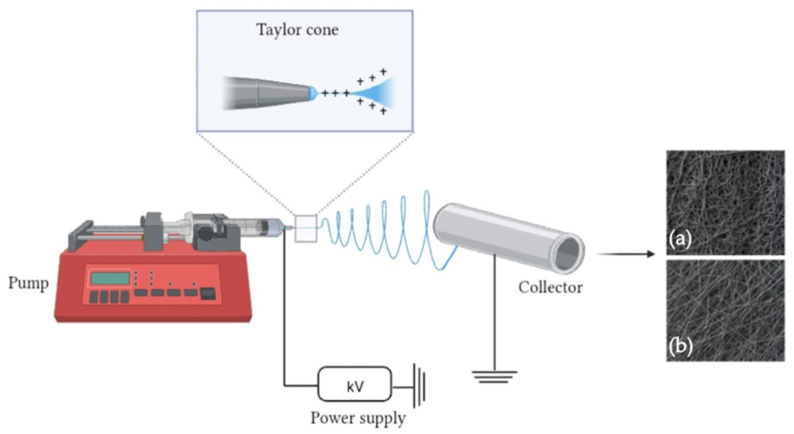
A schematic illustration of the basic setup for blend ES. As an example of the network of electrospun fibers, scanning electron microscopy (SEM) images of (**a**) unaligned and (**b**) aligned membranes have been reported.

**Figure 5 pharmaceutics-15-02725-f005:**
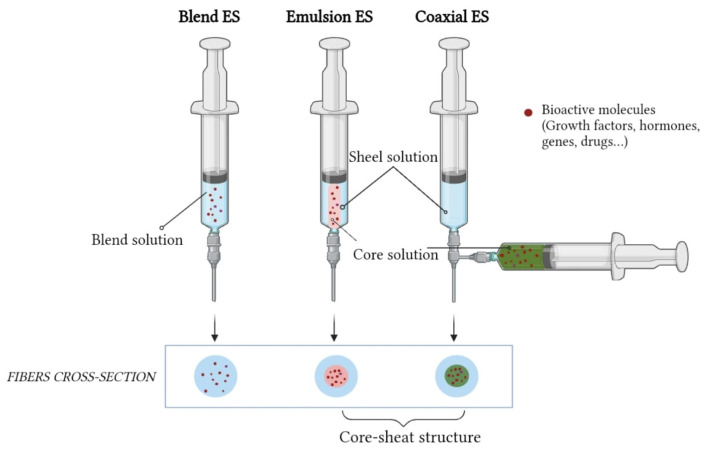
Comparison of blend, emulsion and coaxial ES showing differences in fiber structure.

**Figure 6 pharmaceutics-15-02725-f006:**
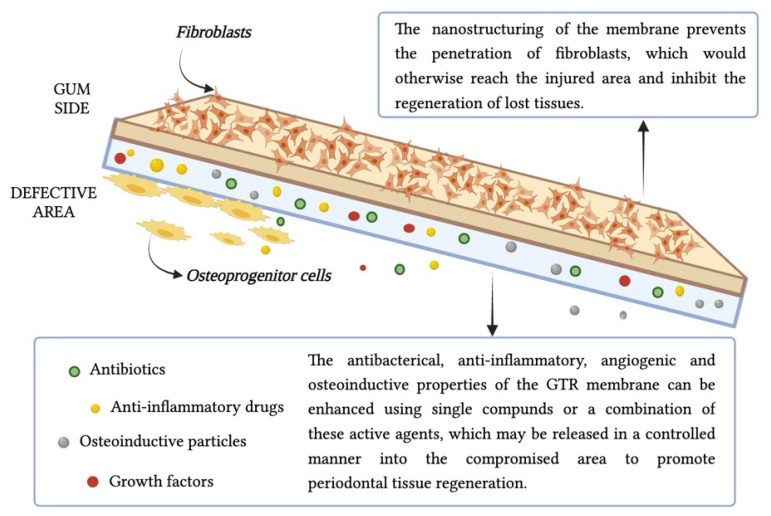
Schematic representation of a multilayer membrane functionalized with bioactive components.

**Table 1 pharmaceutics-15-02725-t001:** Principal characteristics of the most commonly used natural and synthetic materials for the production of electrospun membranes for periodontal regeneration.

Polymers	Advantages	Drawbacks	Ref.
Naturals			
Collagen	Excellent biocompatibility Promotion of cell adhesion	Poor mechanical properties Rapid degradation in vivo	[92]
Gelatin	Integrin-binding sites for cell adhesion Good biocompatibility Low immunogenicity Plasticity Low cost	Poor mechanical properties Fast degradation	[107]
Zein	Good electrospinnability Non-toxic character of its by-products Enhanced cell viability, proliferation, and attachment Suitable for drug delivery	May have immunogenic problem	[108]
HA	High water retention capacity Binding sites for cell adhesion	Inadequate mechanical properties High viscoelasticity High surface tension	[109]
Chitosan	Biocompatible Breakdown by lysozyme Antibacterial, antiplaque activities	Low biodegradation Poor mechanical properties	[93,94,95,110] 12 April 2023 9:58:00 a.m.
PCL	Good mechanical properties Suitable for drug delivery and TE Mechanical strength Biodegradability No acidic degradation products Promotion of osteogenic differentiation	Absence of functional groups, so low cell adhesion and proliferation Slow degradation rate and hydrophobicity	[111] 12 April 2023 9:58:00 a.m.
Synthetics			
Poly (lactic-co-glycolic acid) (PLGA)	Biocompatible and biodegradable Optimal mechanical properties Manageability, flexibility Controllable degradability for GBR applications Suitable for drug delivery	Rapid releases of oligomers and acid by-products may cause significant inflammation reactions in vivo Intrinsically hydrophobic Not optimal for cell adhesion and proliferation	[112] 12 April 2023 9:58:00 a.m.
Polylactic acid (PLA)	Biocompatible Optimal mechanical strength Processability	Slow degradation kinetics Releases acidic degradation products that may induce inflammation Hydrophobicity	[113]
PVA	Water-soluble Absence of toxicity Good mechanical property	Poor cell adhesion High hydrophilicity	[114]

**Table 2 pharmaceutics-15-02725-t002:** Examples of functionalized electrospun membranes for periodontal regeneration.

Compounds	Electrospun Polymers	Outcomes	Ref.
Antibiotics			
MNZ	PCL/zein core/shell nanofibers	MNZ-loaded mats were produced by coaxial ES. The MNA was distributed homogeneously in the core layer of the fibers and incorporation of hydrophobic zein allowed to decrease the initial burst release and prolong drug release period. Notable efficiency against anaerobic bacteria.	[148]
Ampicillin/MNZ	Polylactide fibers	Strong in vitro suppression of oral pathogens such as *A. actinomycetemcomitans*, *F. nucleatum*, *P. gingivalis* was obtained thanks to a synergistic effect of the two drugs.	[149]
Vancomycin	Silk fibroin nanofibers	Negatively charged gelatin nanospheres (GNs) were used in combination with silk fibroin nanofibers, for vancomycin delivery. Biocompatibility and antibacterial effects against *Staphylococcus aureus* were confirmed.	[150]
Amoxicilline (AMX)	PCL nanofibers	Fibrous PCL nanocomposites membranes loaded with various nHAp and AMX contents, to induce possessing osteogenic and antimicrobial activity in vivo.	[151]
NSAIDs			
Ibuprofen (IBU)	PCL nanofibers	The in vivo efficacy of the IBU-PCL membrane was assessed in an experimental periodontitis mouse model, revealing that the IBU-PCL membrane could efficiently and differentially control inflammatory and migratory gingival cell responses and potentially promote periodontal regeneration.	[152]
Aspirin + curcumin		The presence of curcumin and aspirin in the asymmetric membrane enhanced osteogenic potential of the membranes. The results of the animal test showed that the defected area was filled with new bone after 28 days.	[153]
Nanostructure			
Nano zinc oxide–silver biomaterial (nZnO:Ag)	Poly (D,L-lactic acid)(PLA)/poly(lactic-co-glycolic acid) (PLGA) fibers	Unique combination of HA, nZnO, and nAgs reduced cytotoxicity towards osteoblasts, enhanced the biological performance and antibacterial function of the fibers’ coating.	[154]
Magnesium oxide nanoparticles (MgONPs)	PCL	Dual-functional coaxially electrospun membranes by encapsulating parathyroid hormone (PTH) in the core layer and MgONPs in the shell layer were produced. In vivo and in vitro studies demonstrated that MgONPs incorporation has outstanding antibacterial potential, but also significantly prolonged the release of PTH.	[155]

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
