# Peer review of "Recent Advances in Functionalized Electrospun Membranes for Periodontal Regeneration"

_pharmaceutics, 2023, doi:10.3390/pharmaceutics15122725_

Round 1
Reviewer 1 Report
Comments and Suggestions for Authors
The aim of this manuscript is to provide an overview of periodontal regeneration using electrospun membranes, highlighting the use of these nanofibrous scaffolds as delivery systems for bioactive molecules and drugs.
This manuscript shows rich content, providing a deep insight for some works: the study is within the journal’s scope, and I found it to be well-written, providing sufficient information. Even if the manuscript provides an organic overview, with a densely organized structure and based on well-synthetized evidence, there are some suggestions necessary to make the article complete and fully readable. For these reasons, the manuscript requires major changes.
Please find below an enumerated list of comments on my review of the manuscript:
The authors should provide a list of the abbreviations, mentioned in this manuscript.
INTRODUCTION:
LINE 277: As biomaterials are a class of materials, characterized by a unique chemical, mechanical, and biological properties, mainly osteoinductivity and osteoconductivity, which make them suitable and safe to interact with living tissues (see, for reference: Arjunan, A.; Baroutaji, A.; Robinson, J.; Praveen, A.S.; Pollard, A.; Wang, C. Future Directions and Requirements for Tissue Engineering Biomaterials. In Encyclopedia of Smart Materials; Elsevier: Amsterdam, The Netherlands, 2021). In this section, the authors should mention the pivotal role, played by biomaterials in regenerative procedures, due to their specific regenerative properties.
LINE 362: Furthermore, bioresorbable membranes, made of collagen, are also employed, due to their significant functional, and morphological potential which provide the opportunity to examine cellular behavior, as confirmed by structural and ultrastructural studies (see, for reference: Bianchi, S.; Bernardi, S.; Simeone, D.; Torge, D.; Macchiarelli, G.; Marchetti, E. Proliferation and Morphological Assessment of Human Periodontal Ligament Fibroblast towards Bovine Pericardium Membranes: An In Vitro Study. Materials 2022, 15, 8284. https://doi.org/10.3390/ma15238284). This is the major concern of this manuscript: there is a lack of evidence about the contribution of bioresorbable membranes, made for example of collagen, to periodontal regenerative procedures. The manuscript may benefit from highlighting also this topic.
The main topic is interesting, and certainly of great clinical impact. As regards the originality and strengths of this manuscript, this is a significant contribute to the ongoing research on this topic, as it extends the research field on the description of the periodontium and of periodontal disease followed by a summary of some of the current strategies used for the production of electrospun nanofibers for the achievement of periodontal regeneration. Overall, the contents are rich, and the authors also give their deep insight for some works.
Besides, in this manuscript there is a specific and detailed explanation for the evidence mentioned in this study: this is particularly significant, since the manuscript relies on a multitude of methodological and statistical analysis, to derive its conclusions. The results are reliable and adequately discussed.
The conclusion of this manuscript is perfectly in line with the main purpose of the paper: the authors have designed and conducted the study properly. As regards the conclusions, they are well written and present an adequate balance between the description of previous findings and the results presented by the authors.
Finally, this manuscript also shows a basic structure, properly divided and looks like very informative on this topic. Furthermore, figures and tables are complete, organized in an organic manner and easy to read.
In conclusion, this manuscript is densely presented and well organized, based on well-synthetized evidence. The authors were lucid in their style of writing, making it easy to read and understand the message, portrayed in the manuscript. Besides, the methodology design was appropriately implemented within the study. However, many of the topics are very concisely covered. This manuscript provided a comprehensive analysis of current knowledge in this field. Moreover, this research has futuristic importance and could be potential for future research. However, major concerns of this manuscript are previously mentioned: for these reasons, I have major comments for this manuscript, for improvement before acceptance for publication. The article is accurate and provides relevant information on the topic and I have some major points to make, that may help to improve the quality of the current manuscript and maximize its scientific impact. I would accept this manuscript if the comments are addressed properly.
Author Response
Thank you very much for taking the time to review this manuscript. Please see the attachment

Reviewer 2 Report
Comments and Suggestions for Authors
Periodontitis, a persistent inflammatory condition, is marked by the presence of microbes and an intensified immune response, resulting in the erosion of periodontal tissues. This condition can also have consequences for overall systemic health. The restoration of periodontium functionality is a formidable challenge due to its intricate nature. However, there is potential in the utilization of electrospun nanofibrous scaffolds for periodontal regeneration, given their antimicrobial and anti-inflammatory characteristics. The authors aim to investigate the use of electrospun membranes for delivering bioactive molecules to enhance the regeneration of periodontal tissues.
There are several recommendations that can be implemented to improve the clarity and organization of the content.
1. Strengthen and Emphasize Clinical Significance:
Emphasize the clinical importance of maintaining a healthy microbial balance within the human body and the potential consequences of microbial imbalance. This will provide a clear background and context for the discussion of periodontitis and the provision of clear recommendations, ensuring that readers understand the clinical significance of periodontal regeneration and the role of electrospinning technology in this regard.
2. Use Subheadings to Organize and Consolidate Information about Various Tissues within Periodontal Tissues:
Subdivide information about various tissues within periodontal tissues using subheadings. This will help readers understand the details and concepts the author wishes to convey.
Complex Architecture of the Periodontium
2.1 Gingiva: Composition of Periodontal Tissues (including Marginal Gingiva, Interdental Gingiva, Attached Gingiva)
2.2 Periodontal Ligament (PDL): The Tooth's Anchor and Guardian
2.3 Alveolar Bone: Tooth Socket and Protector
2.4 Cementum: Tooth Support and Connection
3. Provide a Microbial Ecological Background:
After a detailed introduction to the complex structure of periodontal tissues, introduce the relevance of the oral microbiome and its role in periodontal health. This can effectively connect periodontal structure with microbiological aspects.
These suggestions aim to enhance readers' understanding of the complex structure of periodontal tissues and their connection to oral microbiology.
4. Organize Information into Several Sections:
To improve readability and promote a better understanding of different aspects of periodontitis, including microbial associations, stages, and systemic connections, organize information into clear sections.
Periodontitis Onset and Progression
4.1 Stages of Periodontitis
4.2 Microbial Factors in Periodontitis
4.3 Systemic Implications of Periodontitis
4.4 Role of Oral Viruses and COVID-19
4.5 Preventive Measures and the Impact on Oral Health
5. Include Visual Aids:
Incorporate graphics or charts to illustrate microbial changes in periodontitis or the systemic connections between periodontal disease and other health conditions. Visual aids can enhance understanding and engagement with the complex information presented in this section.
6. Innovations in Tissue Engineering and Regenerative Therapy:
6.1 Explanation of Tissue Engineering and Regenerative Medicine:
Briefly explain the core concepts of Tissue Engineering and Regenerative Medicine before delving into the specifics.
6.2 Key Components of Tissue Engineering and Regenerative Therapy:
Provide a clear breakdown of the essential components of Tissue Engineering and Regenerative Medicine to aid readers' understanding. Subsections for scaffolds, cellular sources, and regulatory signals can be beneficial.
6.2.1 Scaffolds: The Foundation of Regeneration
6.2.2 Cellular Sources: Challenges and Potential
6.2.3 Regulatory Signals: Guiding Tissue Growth
To provide readers with a more comprehensive understanding of Tissue Engineering, Regenerative Therapy, and the Electrospinning Technique by breaking down the key components and processes involved in these innovative approaches.
7. Functionalization of Electrospun Membranes for Enhanced Periodontal Regeneration:
7.1 Customizing Electrospun Membranes for Enhanced Periodontal Regeneration:
Provide a clear structure to the section, outlining different strategies for functionalizing electrospun membranes to improve periodontal regeneration.
7.2 Specific Functionalization Strategies:
Subdivide this section into subtopics, such as incorporating anti-infective drugs, immune response modulation, incorporating other bioactive agents, and innovative approaches like Gene-Activated Matrix (GAM) and nanomaterials.
7.2.1 Incorporating Anti-Infective Drugs
7.2.2 Immune Response Modulation
7.2.3 Incorporating Other Bioactive Agents
7.2.4 Innovative Approaches: Gene-Activated Matrix and Nanomaterials
These structural improvements can help readers better understand the diverse strategies for functionalizing electrospun membranes and their potential in periodontal regeneration.
8. Conclusion:
Provide a more focused conclusion that summarizes the key points and challenges discussed in the paper and offers insights into future research directions. To provide a concise summary of the key takeaways from the paper and emphasize the need for continued research in the field of periodontal regeneration.
Comments on the Quality of English Languagenil
Author Response

(The authors gave the same response as above.)

Reviewer 3 Report
Comments and Suggestions for Authors
Periodontitis, a chronic inflammatory disease with global implications, is initiated by bacterial microorganisms and an exaggerated host immune response. This condition not only results in the deterioration of the periodontal apparatus but may also exacerbate or contribute to the onset of other systemic diseases. The periodontium, comprising alveolar bone, cementum, gingiva, and the periodontal ligament, necessitates diverse non-surgical and surgical interventions for the restoration of its normal function. Despite efforts, achieving complete regeneration remains challenging due to the disease's etiology and the diverse nature of periodontium components. Addressing this challenge, electrospun nanofibrous scaffolds have emerged as an effective strategy. These scaffolds demonstrate the capability to concurrently exhibit antimicrobial, anti-inflammatory, and regenerative properties. This review offers an in-depth exploration of periodontal regeneration through the utilization of electrospun membranes. It underscores the significance of these nanofibrous scaffolds as delivery systems for bioactive molecules and drugs in the pursuit of restoring periodontal health.
Overall, a significant effort was made to prepare the review, which is quite interesting and unique. This article is impressive for the reviewer and audience of the nanotechnology community as well as material science. This review would show a significant impact on the tissue engineering, nanomedicine, and materials science community. However, the manuscript needs more data to improve the discussion. Moreover, the discussions and introduction are not written clearly and discussed properly. There are many problems with the manuscript as it stands (detailed below) and these need to be addressed before it can be considered further. I thus recommend the paper be reconsidered after major revisions.
Abstract Rewriting: The abstract requires revision to eliminate generic statements and precisely align with the original outlines of the review article. It is essential to maintain a comprehensive focus on the entire topic covered in the review article.
Figures and Copyright Statement: Figures play a crucial role in enhancing the quality of a review article. It is strongly recommended that the authors incorporate additional visually appealing figures in the revised manuscript. Additionally, a copyright statement should be included in the caption of all figures to ensure proper attribution.
Infographics for Reader Engagement: To enhance reader engagement and secure future citations and visibility, the manuscript should incorporate infographics. These visual elements will not only make the content more interesting to reviewers and readers but also contribute to the overall impact of the research.
Introduction Section Enhancement: The introduction section is currently brief and requires substantial improvement. It should provide a clear identification of the scientific problems addressed by the research. Furthermore, an elaboration on biomaterials such as Chitosan, BSA, Gelatin, Zein, and PCL is necessary, supported by recent references (preferably from 2020-2022). The authors have selected nanofibers as the focus, but the rationale for this choice needs emphasis. The introduction should highlight the distinctive characteristics that make fibers a more potent choice for this study compared to other biocompatible and biodegradable systems. Additionally, the authors should underscore the advantages of fibers over alternative systems, citing relevant literature. Exploring biomaterial information (BSA, Gelatin, Zein, PCL, PLA, chitosan, UHMWPE, etc.) in the introduction, referencing recent articles, will enrich the manuscript's quality and captivate reader interest. It is advisable to incorporate pertinent literature quotes to strengthen the manuscript's credibility and relevance. To provide a comprehensive overview of fiber’s multifaceted attributes and applications, it is advisable for the authors to reference recent scholarly works by Ramiro Manuel Velasco Delgadillo, EV Barrera, Fatemeh Ijadi, R.J. Linhardt, and Javier Villela Castrejón.
https://doi.org/10.1016/j.cobme.2022.100436
It is recommended to add a future perspective section to the revised manuscript.
Clinical applications of fibers in dental applications should be reported in a separate section.
According to the corrections, the conclusions may be modified.
Author Response

(The authors gave the same response as above.)

Round 2
Reviewer 1 Report
Comments and Suggestions for Authors
The authors have significantly improved this manuscript.
Reviewer 3 Report
Comments and Suggestions for Authors
No more comments.